# The Role and Efficacy of Creative Imagination in Concept Formation: A Study of Variables for Children in Primary School

**Javier Gonzalez Garcia** [1,*] **and Tirtha Prasad Mukhopadhyay** [2]

1   Department of Music and Performing Arts; University of Guanajuato, Guanajuato 36000, México
2   Department of Art and Business, University of Guanajuato, Salamanca 36766, México
*   Correspondence: jr2000x@yahoo.es

**Abstract:** Children's creative imagination is tested through tasks involving narrative and drawing abilities for participants between the age of 8 and 12 years. The test determines the relative importance of 'narrative' against 'graphic' imagination in interpretive, problem-solving strategies, and also considers how such distinctive functions of the creative imagination could affect 'general' creativity of the child learner. Participants were chosen from designated primary schools in the state of Guanajuato, Mexico. The test on creativity complements facts from observational methodology in a population of mixed Castilian-speaking children. The name of the test is *Prueba de Imaginación Creativa Niños* (2008) or 'Test of Creative Imagination in Children', the Castilian acronym being PIC-N. It comprised four sub-tests: Three designed to evaluate narrative (verbal) creativity, and one for drawing (i.e., graphic) creativity. The first three 'exercises' in the suite indicates (a) fluency, (b) flexibility, and (c) originality in narrative representations, whereas the fourth indexes (d) graphic abilities of the child learner. Results suggest that creative imagination causes variations in specific aspects of creativity, like narrative and graphic improvisation, and also modifies 'general' creativity as understood from the perspective of a developmental psychology of learning abilities in growing children within the defined age group.

**Keywords:** creative imagination; drawing; storytelling; primary education; psychometric analysis

## 1. Introduction to Creativity

*Creative Imagination* appears to be active from an early age and is instrumental in learning and problem-solving strategies. Indeed, such imagination is one of the major ways in which children ascribe meaning and communicate information, attributing new meaning to objects they did not know previously, and to form new experiences from verbal stimuli, images, and also patterns of sound (which was not examined for this project). The archaic research on children's creativity was undertaken by Vygotsky [1], following Piaget [2], but in a manner that predicts the trajectory of the imagination for children's learning and interpretative abilities. Vygotsky's premises constitute some basic laws of developmental psychology for, more specifically, what he was to call "creativity", a term denoting psychomotor processes for producing both mental and physical representations in the material world. Also, 'creativity', as Vygotsky defined it, refers to a correlate of what he called 'imagination'. The latter, in fact, seems to have a more mental and even neural substratum. In this paper, we shall use the term 'creative imagination' to denote integrated neurobehavioral functions that employ imagination [3].

Vygotsky's basic proposition revealed how children could use creative imagination to make sense of this world. As such, the Vygotskian proposition lends itself to later speculations in psychology of children where importance has been laid on the methods of constructing narratives of imagination by children for strategies of interpretation and meaning creation [4]. What is of more immediate

value for this research is the controversy regarding the intrinsic nature of creative imagination. Is children's imagination manifested in organized reproductions of experience already available in real life as Piaget originally claimed, or did it have innate properties that manifested itself in various faculties, like verbal processing or visual memory? Harris' experiments, in this regard, use more of an active developmental angle to explain the process of creative imagination. Harris advocates multiple ontologies that suggest that the imagination functions as a cluster of discrete neurobehavioral faculties, like a multitasking function. Research on children's fantasy undertaken in the 1980s and 1990s underscore how imagination could control a child's understanding of corresponding aspects of reality [5–8]. Assuming that various aspects of the imaginative intelligence could be isolated and tested for their efficacy in the developmental process is perhaps of essence for this project. We consider the singular paradigm of creativity but we are generally inclined to show that facets of language, verbal processing, and visual-imaginative precepts may constitute separate enclaves in children's minds and that it is possible that one or other aspect may be instrumental in the execution of predefined tasks. Interestingly, it should also show that creativity progresses by means of both selective and synchronic organization of elements, and that practical engagement with such elements can lead to a general development for children's interpretative or learning abilities. Among other things, the test packet employed here explains functions of at least two important faculties. Furthermore, observation confirms that such practice could influence development on a long-term basis, for which other temporally spaced tests might be necessary.

Research here indicates that applications of the principle may be effective for teaching and learning at the elementary and middle school levels [9].

We intended to measure how verbal stimuli, generated from reading and storytelling, actually completes and transforms levels of knowledge in the child. We set imaginative tasks to stimulate this sense of meaning and interest in relation to real life problems and to explore why children's fantasy worlds are "so significant and interesting" and see how, as Egan said, we can use "what we learn from this process for educational purposes" [10]. Imagination is a "new psychological process" for the child. It represents a "specifically human form of conscious activity". Like all functions of knowledge, it arises originally from "action" [11]. Since creative imagination can be discovered in the products of creative life, especially of children, we tried to see how elements of representation, by story-telling or drawing, are incorporated in meaningful representations. The task of narrating or drawing objects involves analysis and reintegration, reproducing arrays, and objectivity in complex formations. Hence, the justification for the methodology adopted for the test.

## 2. The Quantitative Framework

There are disputes with the psychometric approach to studies on creativity and creative imagination. Many scholars doubt that it is possible to quantify something as delicate as creative imagination through standardized tests [12,13]. Consistently, in keeping with this line of thinking, Ausubel et al. [14] proposed that creativity was an 'innate capacity'; that is, a particularized and substantial capacity and born with the individual, but it would be difficult to develop it under influences of environment. There are some creative traits, however, that if developed, could turn to function as supportive instruments for the intellect and the personality of a child.

We demonstrate similarly that categorical re-inforcements might actually help children in our designated age group not only to deal with immediate instruments, but also learn to apply creative imagination to such instruments for better alternatives. If proven, intrinsic abilities within creative-imaginative modules may be shown to have positive effects children in the primary school segment. As we said, this proposition is not new in the literature, but our research could probably confirm the presence of variables in the project. We observed creativity in children to hypothesize if certain elements already singled out in the literature are also not instrumental in domain of the tasks for the experiment. We got to confirm that, among other suggested variables, creative imagination involves at least the following steps: (1) 'Encounter', since any process begins with the presentation of a tangible

problem to the individual who engages with it [15]. (2) 'Incubation', during which an individual is continuously inspired by the reality of the problem [16]. (3) 'Enlightenment', corresponding to the results of the process [17]. (4) Self-induced evaluation and verification, in which the participant takes time to check the best options and ideas adopted [18]. We also endorse the profile of the creative subject in terms of nine aspects, among several recognized instruments realized in such authorities as Guilford [19] and Torrance [20]: These are originality, flexibility, production or fluency, elaboration, analysis, synthesis, mental opening, communication and sensitivity to problem, and level of inventiveness [21]. These factors are reduced to more computable categories for the experiment, as are outlined in the section on methodology. Finally, what remains to be said is that these factors, after careful extraction, are set aside specifically for verbal and visual creativity—the two more common sensory functions responsible for semantically valid constructs.

## 3. Existing Models of Creativity: Review and Suggestions

The evaluation of creativity is, thus, a hotly debated topic ranging from skepticism about the possibility of evaluation to asserting that creativity must be evaluated for purposes of improvement and instruction. De la Torre [22] is one author that bets on evaluation. He proposed that creativity requires inter and transdisciplinary approaches, since it has psychological, pedagogical, neurobiological, and sociological connotations [23]. The importance of evaluation of is also defended by authors such as Kaufman [24], who states that creativity is evaluated not only to measure the performance of an individual, but to promote the same type of creativity through stimuli and incentives.

In order to evaluate creativity, different instruments have been employed, all of which have taken into account the three major avenues to its understanding—namely, that of 'creative process' [25–27], and finally, that of quantifying the 'creative product' [28].

We should, thus, draw attention to these attempts in a systematic manner to follow the debate on how attempts have been made to quantify, use, and develop creative practices in real terms. First of all, studies on creative persons are usually aimed at identifying 'personality', 'motivation', 'intelligence', 'cognitive styles', and 'knowledge' that creative people possess or employ [29–31]. Most of these studies conclude that there are certain factors or personality traits that are often associated with high creativity in very different areas. These factors could be increased tolerance for ambiguity, willingness to take risk, self-regulation, interest etc. In this sense, the creative person is described as being more sensitive to problems and information gaps and seems to manifest a need to construct more hypotheses, to investigate and to evaluate problems [26]. Some personality traits appear to be more frequent in people who stand out for their artistic creativity and yet there are others that are more frequently visible in fields of scientific inventions [28].

The other approach to factorial analysis of creative activity has been to consider the conditions responsible for execution of novel tasks. The main models in this group attempts to describe mental operations or phases that a person goes through or experiences during the making of a creative product, such as exploration, incubation, insight, evaluation, etc. This approach also invites categorization of creative thinking in terms that suggest that it is different from non-creative thinking: Such as, for example, the extent to which creative thinking involves conscious, as distinct from unconscious processes, or to what extent creative thinking is the result of external effects such as luck, or of long-term efforts such as persistence and hard work. An example of this process-centered approach is the "Geneplore" model proposed by Finke, Ward, and Smith [31] in which they identify two phases in the creative process: A first of "generation", generating many possible ideas that can solve a problem and a second phase of "exploration", in which the different possible solutions are evaluated to select the most appropriate one. Another well-known approach is the one proposed by Csikszentmihaly [25], which introduces, as we well know, the notion of "flow". We do take cognizance of this model, as well the former one, with their insistence on personality and processual actions.

But the third existing model seems to be more viable, especially for children where personality is less complex, and products of creativity, such as drawing, are easier to deconstruct. Creativity

is a fundamental aspect of cognitive development and is behaviorally manifested, and hence, its measurement depends on variables reflected in the immediate products of psychomotor functions, and this is perhaps more readily visible in children [32]. It is necessary to both evaluate and quantify the behavioral manifests of creativity in order to understand if there are relevant correlations of elements within constituents of those products.

So, as for this creative product model of creative imagination, scholars like Sternberg [33] and Kaufman [24] point out that most creative ideas are characterized by the presence of three components. First, that the creative idea must refer to something different, original, new, or innovative. Second, creative ideas must lead to a quality product. Third, creative ideas will have to be appropriate for the task or problem that is presented. In short, it could be said that a creative product must be novel, of recognizably high quality, and relevant. Studies focusing on the creative 'product' try to determine as to how to judge whether a particular work (poem, musical composition, drawing, etc.) is creative or not. In this sense, some researchers propose to use the consensual assessment technique while estimating the creativity of products or achievements of people or groups [13]. This technique involves judges or experts in an area of knowledge, assigning scores to real products such as a written composition, a poem, or a drawing. We would like to argue in favor of the fact that creativity should always be evaluated in such concrete domains. An evaluative-quantitative approach centered on the product has the potential of being more objective, although it would have the drawback of not allowing us to know how the subject reaches the end product. It also makes it possible to identify subjects that have been remarkably creative, but it does not make it easier to identify subjects whose creative potential are yet to be developed [34].

The fourth factor, the pressures of the environment, emphasizes how the social environment, the models to which one is exposed and the cultural values and attitudes that surround us contribute, sometimes to a great extent, to foment or inhibit our creativity.

## 4. Creativity Testing in Children

Torrance [35] has made great contributions to the study of creativity; he became interested in things that can be done before, during, and after a lesson to increase creative thinking. He indicated that creativity is a process that is expressed in changes discovered, is a capacity that can be developed and, in children, it is something that is confirmed through their productions, such as stories, fantasies, and drawings. In addition, he designed a test to evaluate the four basic skills that reflect creativity: Namely, fluency, flexibility, originality, and elaboration. In the most recent version of his test published in 2008, he proposed some changes to the proposed indicators: Fluidity here refers to the ability to produce a large number of ideas; originality now involves, for him, the ability to contribute ideas or solutions that are far from the obvious, common, or established; elaboration appears as the aptitude of detailing ideas; finally, again, titles for creativity refer in the latest version of the analysis to the ability to generate ideas that capture the essence of drawings and sensory reproductions. Torrance also speaks of the ability to generate original ideas, with intense images and details in addition to the stimulus.

The *Torrance's Creative Thinking Test* [36] is based on Guilford's theory of intellect [37] and is a useful tool for evaluating both quantitative and qualitative aspects of divergent thinking, especially creative products [38]. This instrument consists of a group of useful tests to evaluate the creative process as a whole and also the specific skills that define it. The figurative expression section of Form 1A evaluates the level of imagination in constructs like drawings and computes the products of the three following activities: (a) Composition, (b) finish, and (c) parallels. The definition of creativity is far from reaching an unanimity among members of the scientific community, but despite the diversity of perspectives, most authors agree that it is a complex and multifaceted construct. Davis and Wechsler [39] and Vendramini and Oakland [40] validate the Torrance test for the evaluation of divergent thinking; these authors recognize the accuracy and validity of the assumptions regarding the structure of the novel image. Most other similar studies report on the structural validity of the dimensions assessed in the subtests of this test (fluidity, flexibility, originality, and elaboration) and even establish high correlations

between them. The works of Guilford [19] and Torrance [35] set a milestone in the study and evaluation of creativity based on psychometric and factorial perspectives.

Empirical evidence supporting the test proposed by Torrance [35] had its beginnings with the 1959 analyses of high school students where the three of the components of his test—fluidity, flexibility, and originality—were extensively confirmed and agreed upon as being the best predictors of creative achievement. Five years later, Torrance resent a questionnaire to the entire population that participated in the research of 1959 and found that all predictors of creativity were significant at a level of 0.01. In order to examine the relationship between the various indicators, a later study used Plucker's structural model in which it was argued that intelligence also has a positive effect on Torrance's test [41]. Increments in the four factors—fluidity, flexibility, originality, and elaboration—when evaluated by judgments of experts, reinforced the predictive validity of the test. In Plucker's research, data were examined under a Pearson product-moment correlation and a positive relationship was found between the indicators of creativity and the criteria of intelligence in creative achievement. Various analyses such as Plucker's have confirmed that the Torrance test is highly predictive of creative production in each of its indicators. Research of Terman and Oden (1959), Bloom (1985) [42], and Torrance (1993) with highly intelligent and talented individuals emphasized the critical role of personality factors, opportunity, experiences, and other environmental aspects that could play a role relevant to the development of creativity. However, it is very clear that there are other additional factors that can help or prevent the recognition of creative characteristics.

## 5. The Socio-Educational Context

Several authors such as Sánchez, García, and Valdés [43] have indicated that in a country like Mexico, there are no valid or reliable instruments of measuring children's creativity. Therefore, it was considered indispensable to validate Torrance's creative thinking test for a sample of Mexican students and to confirm psychometric properties of creativity in the population analyzed. Creative development provides a path of maturation in which creative activity manifests itself at different levels and in different ways. The present research aimed to analyze the incidence of imaginative strategies solely for the narrative and drawing tasks assigned to the children of the fifth grade in primary schools in Guanajuato, Mexico. The application consisted of a set of imaginative strategies in narrative and drawing tasks for an experimental group, and also for children not subject to the program, or the control group.

The assumptions of Torrance [35] and the later Torrance (2008) were considered as the starting hypotheses. Individuals who exhibit high levels of creativity have a greater potential to benefit from creative educational experiences and could be compared to groups of children with high and low levels of creativity [34] as is also evidenced in the research of Olveira, Ferrándiz, Ferrando, Sainz, and Prieto [44]. As for the only specific precedence, Soto and colleagues [45] already analyzed the construct validity of the Torrance creative thinking test with a sample of 500 students from a primary school located in an urban-marginal zone of the Iztapalapa delegation in the Federal District of Mexico.

Soto and his colleagues found that children with high scores showed an increase in graphic (visual) creativity and also originality, elaboration, fluency, better title construction, and closure. Children with low scores demonstrated increases in visual creativity and indicators of fluidity, originality, and titles. It is important to note that, although significant improvement in total creativity was achieved in both groups, a separate analysis was done for the group of highly creative children. Positive changes were observed in all the indicators, while in the group with low observed creativity, increase was recorded only in fluency, originality, and title-writing. Likewise, a decrease in the values of elaboration and closure was found for the same by Soto and his colleagues. Differences in individual parameters support the need of considering a variety of indicators for evaluating creativity, since the scores exhibited significant increases in both intragroup and intergroup creativity, as well as in each one of the indicators discussed. Different aspects of the same construct of creativity may show a certain independence from one another and this may be a desirable quality of the instruments of measurement.

Likewise, performance tests, questionnaires, self-reports, and tests involving subjective judgment have been used to assess creativity in children from other Latin American countries [23,46,47]. The tests of performance or skill that quantify the creative process, primarily of divergent thinking, are the tests that have focused most on the psychometric factors of creativity. In this sense, López-Martinez and Navarro-Lozano [47] talks about tests of divergent thinking that are elaborated from Guilford's SOI test. We also refer to Perez and Avila [48], whose tests are based on Guilford's multifactorial theory of intelligence, which states that creativity is not an independent dimension, but is integrated in contexts of many cognitive functions. Performance tests in Spanish made considerable use of versions of the PIC instrument. In addition to performance tests, the *PIC* has related assessment tools: Self-report, questionnaires, inventories, and scales that evaluate the characteristics of members in the sample. But these tests have limitations and are specific to the objective of research. Based on what has been said, this study aims to analyze if there is a statistical relationship between the scores obtained in narrative and drawing manifests of creative imagination and, unlike Ramirez or Lopez's study of additional variables, aims to be a comparative analysis of factors that are agreed upon in the literature. The *PIC-A* tests is based on the Guilford test battery, the turtle test, and latest versions of Torrance, and may be explained in detail for purposes of arriving at conclusions regarding merits of relative factors in creativity and also their synergic relationship to the broad question of how creative imagination functions as a system.

## 6. Methodology

The *PIC*, or *Prueba de Imaginación Creativa Niños* [49], was applied to count creativity for the experimental group (as well as the control group). The *PIC* may be translated as the 'Test of Creative Imagination in Children'. Generally, the *PIC* may be defined as a test of divergent thinking that evaluates creativity by examining how individuals use creative imagination in their creative representations. There are three versions of the *PIC*, for the levels of children *PIC-N*, adolescents *PIC-J,* and adults *PIC-A,* respectively, but all three versions have a similar structure of factorial measure for both verbal creativity and visual or graphic creativity.

Its application can be either individual or collective; in our case, the *PIC* was applied to participants individually. It also measures a 'general' creativity index for each participant. The scope of the experiment was restricted to school children between the third and sixth year of their primary education (with ages between 8 to 12 years). In terms of duration, the application could be variable, although for our experiment, a time-frame of approximately 40 min was chosen. For this duration, the *PIC* could be used to measure things like the ability to: (a) Formulate hypotheses (Exercise 1), (b) to think in an unconventional way (Exercise 2), (c) to exploit imagination and fantasy (Exercise 3), and (d) to employ, specifically, the graphic or imagistic imagination (Exercise 4). Results for the *PIC* are expressed in percentile for each variable studied and for each course of activity. Finally, we need to say that the materials for the test application was comprised of a test manual, an exercise copy, and a correction booklet.

## 7. Sample

The sample consisted of 300 children, 150 of them belonging to the experimental group (75 boys and 75 girls), and 150 corresponding to the control group (75 boys and 75 girls) from two public schools in Guanajuato, Mexico. The ages of the children ranged from 8 years to 12 years; students had to meet the following selection criteria: (a) Children could not have repeated or failed courses, (b) they could not have taken art classes (like drawing or painting) or literature classes, in which parents or guardians in charge might have influenced them in one of these two areas by demonstrating precepts of imaginative instruction and interpretation. These aforementioned variables, namely those of art or literature, were considered to restrict any prior intervention of factors that could have either affected or strengthened children's awareness or employment of creative imagination in some way or the other. Children who had this prior stimulus may have had higher scores. To evade the intervention of these

variables, we had to make sure that the children population were equivalent in both the experimental and control groups.

## 8. Instruments

The *PIC* 2004, as we said, evaluates narrative creativity and graphic creativity to obtain a result of general creativity for the imagination of children between 8 and 12 years. The 4 "tests" or "exercises" are used to count four different facets of creativity originally suggested by Torrance (1966): Namely, fluidity, flexibility, elaboration, and originality, both in their narrative and graphic aspects. It offers scores on each of these facets and gives then a sum of both narrative and graphic aspects to render a total score of creativity. It may be characterized as a simple instrument, easy to apply or readjust with the help of guidelines given in the test manual. The term "game" is used on the front cover of the tests in order to minimize the impression of an evaluation or examination for participating subjects, and to obtain results of the application in an environment free from strict regulations since this encourages the ludic tendencies of creative subjects. The first game evaluates fluidity and flexibility with the help of an image in which children must write something by using given phrases to represent all the ideas that they consider may be passing through their minds. In the second, fluidity, flexibility, and originality are evaluated through a score for arrangement of one or several rubber tubes of different sizes. The third test poses an unexpected situation for the young participants, which is that they would need to imagine what would happen under implausible conditions, like in a situation in which a squirrel would suddenly turn into a dinosaur. Finally, four incomplete drawings are presented in the fourth game. Here, children had to complete each inchoate figure by drawing out something no one has thought of before.

A pre-test and post-test were applied, for evaluating fluidity, flexibility, originality, and elaboration according to the following criteria:

- Fluency: Evaluated in games 1, 2, and 3. One point is registered for each proposed idea. Repeated and non-pertinent answers are eliminated. To obtain score the total points for the individual games (1, 2, and 3) has to be added and placed in the corresponding box.
- Flexibility: This is evaluated in games 1, 2, and 3. Again, one point is registered for each different category. The number of categories is counted. To obtain the total, points from individual games (1, 2, and 3) must be added and then placed in the corresponding box.
- Originality: This is evaluated in games 2, 3, and 4. Scores are obtained by multiplying the coefficient that appears on the right, to obtain the total; similarly, all points must be added and placed in the corresponding box. This score is given on a range of 0 to 3. In game 4, originality is evaluated graphically by comparing the given answers of each drawing with the tables of originality that are provided. A score of 0, 1, 2, or 3 has to be assigned depending on the matching statistical frequency of the response.
- Elaboration: The answer given to each drawing was assigned a score of 0 to 2 according to the amount of details in it.

The application of the test was carried out individually, since it was assumed that the creative imagination is a process that is strengthened individually.

## 9. Description of Experimental Procedure

To achieve the objectives set out in this research, which is to determine how creative imagination augments learning and interpretive abilities, we examined an experimental group with pretest-and-post-test measurement, with a control group.

This measurement was made to determine the effects of a set of imaginative strategies (independent variables) on the dependent variable (creative imagination). In order to increase the internal validity of the study, intervening variables were controlled, thus excluding variables such as training and repetition.

## 10. A Detailed Report of the Pre-Test Activity

(1) A pre-test procedure was carried out, in order to ascertain the initial level of creative imagination in participants of both groups. This was done on an individual basis, in the same school, and on the same day.

Teachers and parents were informed of the objectives of the study and were solicited for confidentiality. The families of the students gave authorization to carry out the study.

Prior to the application of the program, the group classes that were to constitute the experimental group where the program was applied, and the group that was to constitute the control group, in which only the *PIC* was applied, were randomly selected. The test was applied before and after the development of the program in the experimental groups. The program was carried out in two centers.

The *PIC* pass is made during class hours and in a classroom where students are alone and separated from their peers. It is a sunny classroom, well ventilated, and noise-free, following the original instructions of the test manuals [49]. The application is made by the school counselor of the center.

Each child received a copy of the application booklet. During the application, the children had a varied material to paint (pens, waxes, markers, pencil sharpeners, erasers, etc.). The instructions were adapted to the children's age and comprehension skills. The applications were collective respecting the distribution by classrooms, in a single session.

The test is applied at the beginning (pretest) and at the end (post-test) of the program development to both the control group and the experimental group.

(2) Subsequently, it was applied to the strategies of the creative imagination that contained new activities in the experimental group. The first strategy for increasing fluency and flexibility required four activities (Guilford circles, mystery problems, brainstorming, other uses of everyday objects). The other strategy for incremental originality and elaboration was shaped by five activities (verbal analogies, relaxation and imagination, the best animal in the world, improvement of designs, and the machine without use), which sought to increase the participants' creative imagination through indicators raised by Guilford (1950) and confirmed and developed by Torrance. Once again, these are fluency, flexibility, originality, and elaboration.

## 11. Requirements and Exercises

The test consists of the accomplishment of exercises of complementation of drawings or graphs previously designed and proposed so that the students complete them; they must provide all the necessary ideas to make the drawing interesting. The test evaluates the creativity from the use that the individual makes of his imagination. It consists of four exercises: The first three assess verbal or narrative creativity and the fourth evaluates graphic creativity.

**Exercise 1**. In this exercise, from a situation that is reflected in a drawing, the individual has to write everything that could be happening in the scene. The presented stimulus varies according to the version of the *PIC* in question: In the *PIC-N*, a boy opening a chest; in the *PIC-J*, a boy and a girl in a lake; and in the *PIC-A*, an ambiguous scene is presented in the street, in which several characters appear. This game allows the person to express their curiosity and imagination and has been included to explore the ability to formulate hypotheses and think in terms of the possible. The test allows the expression of curiosity and speculative attitude; the ability to go beyond the information provided by the stimulus, by posing different possibilities with respect to what is imagined to occur on the scene.

**Exercise 2**. This exercise consists of a test, in which a list of possible uses of an object must be elaborated, according to Artola et al. [49]. In this case, it is: "Uses of a rubber tube"; in this exercise or subtest, the stimulus presented is the same in *PIC-N* as in *PIC-J* and *PIC-A*. This test is included as a measure of the ability of individuals to free their mind and think in an unconventional way; allows the evaluation of the "redefinition" of problems, that is, the ability to find uses, functions, and applications different from the usual ones; to quicken the mind; and to offer new interpretations or meanings to

familiar objects, to give them a new use or meaning. Subjects can use the number and size of tubes they want. As in the first exercise, there is a list from 1 to 38, and it starts with an example.

**Exercise 3.** The exercise proposes subjecting subjects to unlikely situations. The situation presented varies according to the version of the *PIC*. In *PIC-N*, the situation is as follows: "Imagine what would happen if each squirrel suddenly became a dinosaur"; *in* the *PIC-J*, the situation is: "Imagine what would happen if the ground were elastic"; *in* the *PIC-A*: "Imagine what would happen if we did not stop remembering". The exercise evaluates the fantasy aspect of the imagination. This way of thinking seems very important in creative behavior. This exercise identifies the capacity for fantasy and the ability to handle unconventional ideas, which the subject probably would not dare to express in more serious situations, as well as openness and receptivity to novel situations. It is interesting how the test allows for the evaluation of the ability of "penetration" of the subject or ability to delve into experiences. Some of the consequences of the presented situation are obvious and simple to discover, while others, more remote, require a deeper study of the matter.

**Exercise 4.** This exercise is a graphic imagination test, inspired by the items of the Torrance Test (1966), according to Artola et al. [49]; in it, the subject must complete four drawings from a given stroke and put a title to each one of them. According to the authors, the incomplete figures used in game four have been selected, after presenting several figures to a sample of people considered as very creative (included in a program for gifted individuals), selecting the four that are most suggestive for them. The only premise, before starting the test, is to ask them to try to draw a picture that no other person could imagine; equally, the answers must give all the ideas necessary to make the drawing interesting.

## 12. Post-Test Assessments

(1).   Nine activities were planned with three sessions per week. These activities took into account everyday situations and objects that allowed participant children to express their imagination through free expressions and reflecting them in a concrete way. Especially, narrative and graphic aspects were stimulated given that both aspects are exercised in different areas of activity in school.

(2).   After the intervention, the post-test evaluation was performed for both groups individually. The objective was to analyze the impact of imaginative strategies in the narrative and the drawing reflected in the context of a significant increase of score in the experimental group.

(3).   After the post-test evaluation, an open interview with the teachers was used to compare the results obtained and to develop a more in-depth discussion of the results of the intervention. (Appendix A).

## 13. Results

Exercises 1–4, in their individual and combined applications, yielded the following results for the four conditions underlined by Torrance [35] and developed in contexts of schoolchildren in Mexico by Ramirez and colleagues [45], and also for investigative paradigm among primary school children in Chile by Lubart [38]; Lopéz and Navarro [23] indicate a *p*-value < 0.01 for general creativity for a comparative analysis of experiment and control groups in these categories.

Through Tables 1 and 2 we can observe the evolution of results before and after the program of nine activities for creative development. Through the evolution of means and standard deviations, the impact of the imaginative strategies in the narrative and the reflected drawing is observed in the context of a significant increase in the score in the experimental group.

**Table 1.** Mean and standard deviation calculated on EXCEL for pre-test variables of creativity for the individual components suggested in the literature following Torrance (1966).

| Variable | Group | Median | Standard Deviation |
|---|---|---|---|
| Narrative Fluidity | Experimental | 19.3 | 3.42 |
| | Control | 20.2 | 3.49 |
| Narrative Flexibility | Experimental | 17.6 | 3.31 |
| | Control | 18.6 | 3.43 |
| Narrative Originality | Experimental | 11.2 | 2.91 |
| | Control | 11.5 | 2.87 |
| Graphic Originality | Experimental | 6.1 | 2.03 |
| | Control | 5.9 | 2.19 |
| Special Details | Experimental | 0.2 | 0.12 |
| | Control | 0.2 | 0.11 |
| Title | Experimental | 2.1 | 1.03 |
| | Control | 2.2 | 1.05 |
| Color and Shade | Experimental | 0.3 | 0.07 |
| | Control | 0.4 | 0.13 |
| Elaboration | Experimental | 2 | 0.4 |
| | Control | 1.8 | 0.45 |

**Table 2.** Standard deviation in post-test variables for individual Torrance components of narrative and graphic creativity for individually tested experimental and control groups.

| Variable | Group | Mean | Deviation |
|---|---|---|---|
| Narrative Fluidity | Experimental | 25.4 | 3.82 |
| | Control | 23.5 | 3.19 |
| Narrative Flexibility | Experimental | 21.2 | 3.05 |
| | Control | 19.1 | 3.13 |
| Narrative Originality | Experimental | 15.3 | 3.41 |
| | Control | 14.5 | 3.07 |
| Graphic Originality | Experimental | 6.1 | 1.92 |
| | Control | 5.9 | 2.01 |
| Special Details | Experimental | 0.2 | 0.24 |
| | Control | 0.2 | 0.15 |
| Title | Experimental | 2.4 | 0.93 |
| | Control | 2.3 | 1.12 |
| Color and Shade | Experimental | 0.3 | 0.07 |
| | Control | 0.4 | 0.13 |
| Elaboration | Experimental | 1.6 | 0.41 |
| | Control | 1.7 | 0.42 |

Table 3 shows results of inferential analysis obtained from the differences following interventions related to each one of the variables, namely of verbal and figurative creativity. In the first column, the mean change score is presented; in the second, standard deviation; while in the third column, the significance of this difference is presented. In the follow-up test, the test scores of the TCTT and the PIC-J are not comparable to each other at intra-group level because they are measured on different scales of measurement. For this reason, only the differences observed at the intergroup level are analyzed (Table 3).

**Table 3.** Mean, standard deviation, and ANOVA results of intergroup differences in the creativity of the follow-up test (*PIC-N*).

| Variable | Group | M | SD | P |
|---|---|---|---|---|
| **Narrative Creativity** | Experimental | 52.5 | 7.02 | 0.009 ** |
| | Control | 46.3 | 6.51 | |
| Narrative Fluidity | Experimental | 27.1 | 3.82 | 0.072 * |
| | Control | 26.6 | 3.19 | |
| Narrative Flexibility | Experimental | 15.4 | 3.05 | 0.0001 *** |
| | Control | 15.9 | 3.13 | |
| Narrative Originality | Experimental | 11.8 | 3.41 | 0.005 ** |
| | Control | 12.2 | 3.07 | |
| **Graphic Creativity** | Experimental | 13.46 | 4.42 | 0.001 * |
| | Control | 12.12 | 3.97 | |
| Graphic Originality | Experimental | 5.6 | 1.92 | 0.72 |
| | Control | 5.5 | 2.01 | |
| Special Details | Experimental | 0.17 | 0.05 | 0.002 * |
| | Control | 0.11 | 0.05 | |
| Title | Experimental | 2 | 0.93 | 0.42 * |
| | Control | 2.2 | 1.12 | |
| Color and Shade | Experimental | 2.4 | 0.07 | 0.002 * |
| | Control | 2.2 | 0.13 | |
| Elaboration | Experimental | 2.4 | 0.41 | |
| | Control | 2.3 | 0.42 | |
| General Creativity | Experimental | 65.4 | 7.88 | 0.005 ** |
| | Control | 60.3 | 6.62 | |

Note: *** $p < 0.001$. ** $p < 0.01$ * $p < 0.05$.

The results obtained from the narrative and graphic creativity in the monitoring test at the nomothetic level show significant differences between the EG and the GC in most of the creativity variables (Table 3). Specifically, the development of GE creativity is significantly greater than that of CG in narrative flexibility, special details and graphic creativity ($p < 0.001$); in the elaboration, the narrative creativity, the graphic creativity and in the general creativity ($p < 0.01$) and; in graphic originality ($p < 0.05$). The development of narrative fluency and the title of the GE is greater than that of the CG, but the difference between one and the other is not significant ($p > 0.05$).

A comparison of means was carried out using the Student's *t*-statistic, in order to evaluate the level of significance achieved after applying the strategies to the control group. Similarly, Pearson correlation and linear regression between each indicator were established, and the overall result for creative imagination of the experimental group was determined to verify the degree of association between them. For this analysis, we used the statistical program SPSS for Windows. The following graphs show the results according to the following criteria. In the X axis, there are six ranges that contain the direct scores distributed in intervals: Where the range of the number of is greater, the greater the score. The Y-axis represents the relative frequencies, which indicate the percentages of the absolute frequencies of subjects grouped in each interval. Subsequently, the level of significance of the totals in both groups are analyzed with the Student's T, which is established from the midpoint of the six intervals, in which the direct scores obtained by the subjects are distributed. Finally, we establish the Pearson correlation in the experimental group between the indicators of the narrative and graphic creativity in front of the general creativity as evidence of influence. The following is a summary of the results of the open research.

In Figure 1, the narrative creativity dimension, results were obtained in the first four ranks for the control group, with a higher percentage in the third. In the experimental group, however, no results appear in the first column, and show an increase in the fourth column, reaching maximum representation in the last one. This points to an increase in scores for the experimental group as opposed to the control.

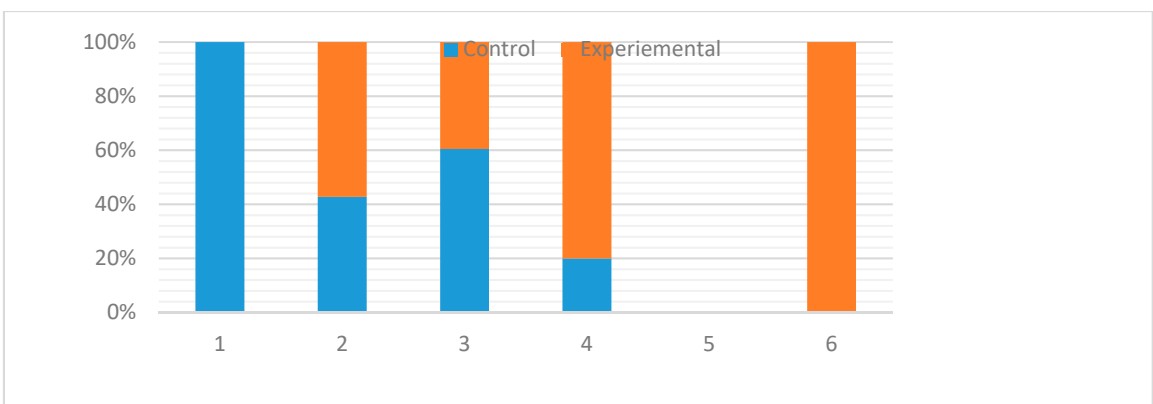

**Figure 1.** Narrative Creativity.

As shown in Figure 2, in the context of graphic or image-based creativity, the experimental group achieved its highest performance in the third range, followed by in the fourth one where its highest percentages are noted. This, as well as what we see is the fact that it obtained high percentages in the last two columns, unlike as in the control group, shows its highest percentages in the first three columns, standing out in the second, and without showing results in the last two.

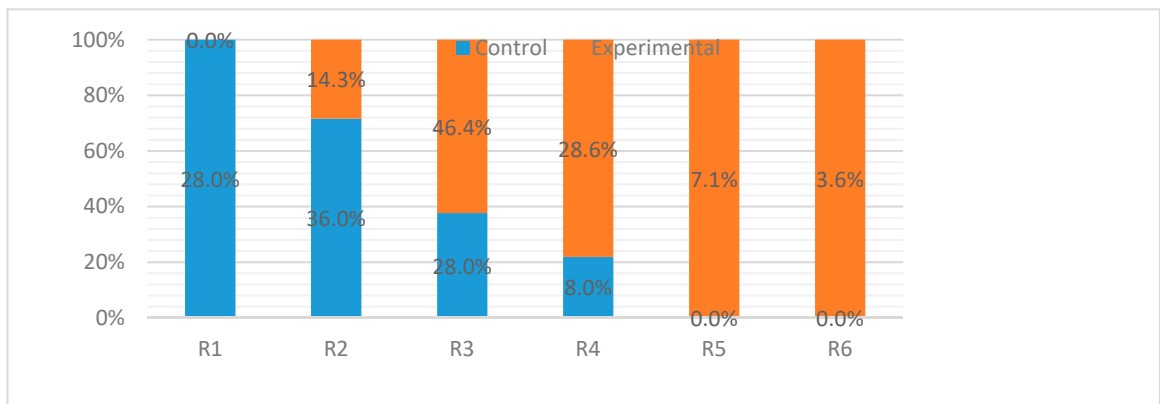

**Figure 2.** Graphic creativity (post-test).

As for general creativity (Figure 3), the control group scored in the first three ranges with their highest score in the third, unlike the experimental group that showed results from the second to the sixth range, but with the exception of the fifth.

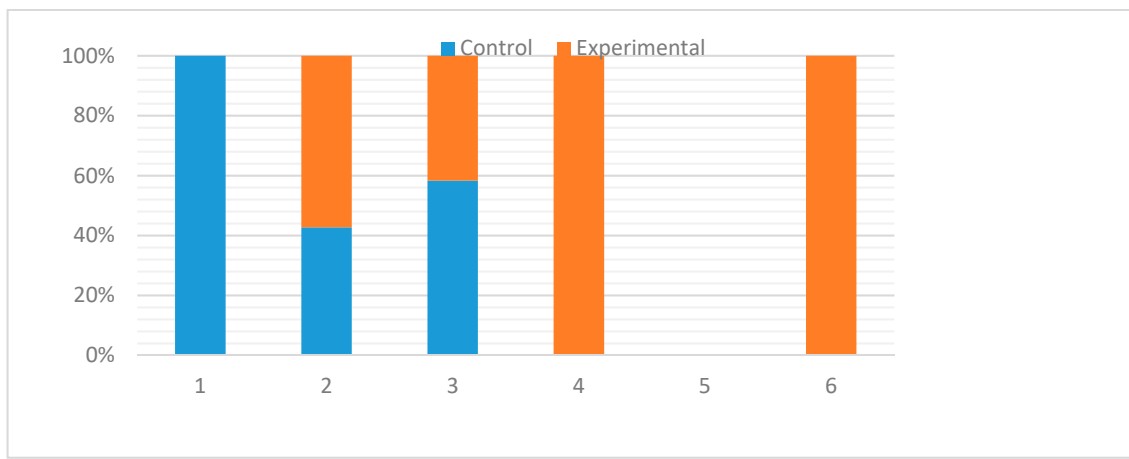

**Figure 3.** General creativity post-test.

As observed, the correlation is significant at the 0.01 level between the narrative dimension of the test and that of overall or general creative imagination. This result is explained by the fact that the strategies designed have a high component of activities related to narrative representation: Seven activities out of nine focus on narrative creativity. This means that the component that had the highest incidence in relation to the PIC test was narrative creativity. It seems that narrative creativity has a high correlation with fluidity, flexibility, and narrative originality (see Table 4).

**Table 4.** Correlation Creativity Narrative to General Creativity.

|  |  | **NARRATIVE** | **GENERAL** |
|---|---|---|---|
| NARRATIVE | Pearson Correlation | 1 | 0.994(**) |
|  | Sig. (bilateral) | - | 0.000 |
|  | N | 300 | 300 |
| GENERAL | Pearson Correlation | 0.994(**) | 1 |
|  | Sig. (bilateral) | 0.000 | - |
|  | N | 300 | 300 |

Note: *** $p < 0.001$. ** $p < 0.01$ * $p < 0.05$.

Unlike the previous result, the graphic dimension does not have a high correlation with a general creative imagination scale (Table 5). The results suggest that graphic originality and graphic design do not correlate more closely with total overall creativity (Appendix A). This means that the items of the PIC test, as well as the strategies employed, support the changes of creativity in terms of narrative rather than graphic creativity activity. However, as suggested in the discussion, the experiment demonstrates evidence of the importance of writing and narrative in general creative imagination.

**Table 5.** Correlation of graphic creativity to general creativity.

| **CREATIVITY** |  | **GRAFIC** | **GENERAL** |
|---|---|---|---|
| GRAPHIC | Pearson Correlation | 1 | 0.135 |
|  | Sig. (bilateral) | - | 0.493 |
|  | N | 300 | 300 |
| GENERAL | Pearson Correlation | 0.135 | 1 |
|  | Sig. (bilateral) | 0.493 | - |
|  | N | 300 | 300 |

## 14. Post Test Qualitative Assessment

It is important to note that a complete evaluation of children's creative capacities requires attending to other environmental, academic, familial, and social factors that may influence the overall development of the child. This implies including qualitative information from observation scales performed by teachers or from interviews with children, parents, and teachers, or taking quantitative measures of attitudes and interests, peer nominations, teacher nominations, supervisor evaluations, product judgments, self-reports of activities, or creative achievements [50]. Taking as reference the Renzulli–Smith student characteristics rating scale and the Monterde primary school pupil observation scale, we followed up on open interviews and discussion groups with tutors and teachers that attended the performance of the sample of students. The objectives were several: To have a closer look, although approximately, at their habits outside the school; inserting reflections on creativity in a wider framework of knowledge of the people who contribute to the children's formation and on the places that host them. We met and listened to the people who lived with the sample of children, as a way to establish a relationship of mutual trust and respect and to approach the peculiar reality of schools. Interviews

were conducted on the same school day, in which tests were applied to children, during recess or at a teacher-free time.

Tutors described boys and girls as generally restless and cheerful, and healthy in the broadest sense of the word. They defined them also as very autonomous in their daily chores. Most children come and go by themselves to school, are accustomed to go shopping from the age of seven, have their gang of friends with whom they go out, and know to organize their games.

In the opinion of the tutors, the areas in which children showed more creativity are organization of games using resources offered by the environment, the resolution of conflicts between themselves, and their crafts. The tutors agree that the environment in which they live offers endless resources for the promotion of their creativity. This, in addition to the freedom granted by their parents, helps them to become more aware of their skills and interests. Tutors considered autonomy as a fundamental aspect of the development of creativity. In their environment, children moved with confidence and responsibility; outside of it, during cultural visits for example, they seemed to lose their safety and sometimes appeared disoriented. In this case, the difference between those who are accustomed to going out with the family and moving in different contexts and those who are not, were noted. In general, they do not show special fears but are cautious and respectful of the unknown. There are no evident problems of discipline. Children manifest in general self-control and respect towards the school environment and the tutors.

The group of teachers who either attended or took care of the children reported positive attitude and motivation after the experience generated by this research project. Regardless of the disciplinary field in which they are found, the after-effects of artistic education can be adjusted to all educational processes. A majority of the teachers agreed that transdisciplinary performances were generally conducive to civic and ethical education, English, mathematics, and Spanish. A second group endorsed transdisciplinary plastic arts education for better development of precepts in civic formation and ethics, mathematics, and natural sciences. Likewise, this group of teachers expressed that they observed creative thinking skills in their fourth-grade students, which are similar to those that formed part of the conceptual framework of our research, especially to fluidity of thinking, originality, and elaboration. The rest of the qualities stated by the teachers have little relation to the qualities that reflect creative thinking abilities addressed in the conceptual framework of this research.

Most teachers agree that creativity does not consist of the perfect use of a technique to perform an artistic work, because they recognize the need of creativity to solve problems of daily life as well. Creative attitudes are often observed in students, in formats such as socializing and acquainting with new and unpublished objects or creations, followed by indications that they relate to knowledge from different fields to express their opinion on a topic. It includes production of novel hypotheses and questions on introductory topics, in the same way they express gracious analogies about everyday situations. It is logical to assume that many teachers find it very difficult to drive and follow up on experiences generating creative thinking skills in their educational plans if they themselves had no chance to experience them during their training processes in their respective training faculties, i.e., it is not enough to learn from the concept of memory or copy the characteristics of such thinking. It is necessary to make the activity existential [44].

Only two teachers were trained in concrete subjects of education and artistic appreciation, and a great majority are unaware that children could be educated through the creative imagination. Moreover, those who know him do not know how to do it. This awareness should lead to teachers using innovative strategies in the classroom, and to believe in education, because children can be educated in other ways.

## 15. Discussion of Results

The implementation of the set of strategies confirmed the role of creative imagination in generation of intelligible narratives and a smaller degree of its presence in graphic or imagistic solutions. While selecting the activities that formed the strategies for the increase of the creative imagination, four indicators were followed: Fluidity, flexibility, originality, and elaboration. Activities should involve

criteria for the strengthening of each indicator, which, as a whole, would allow development of the creative imagination.

Children could have unique ideas that could be positively valued and could arguably provide them the confidence to freely express their ideas without the fear of being told that they are wrong.

Children who received the treatment showed receptivity and seemed to enjoy the activities. Creative imagination is a capacity that can be enhanced through the implementation of relevant strategies. Considering that both groups were equivalent with respect to the results of the pre-test, it was clearly noticed that the experimental group, while receiving the treatment, reached significantly higher levels of score in general creativity. Achievement of this increment means fluidity, flexibility, and originality are causative factors for the process. Indicators of narrative creativity demonstrated a high correlation compared to general creativity. This means that narrative indicators are relevant for the strengthening of the creative imagination. On the other hand, when originality and graphic activities are compared, indicators did not appear to reach superior significance levels, but demonstrated low positive correlations.

This result, which is still provisional, allows us to believe that despite availability of novel resources in direct acts of manufacture (involving things like unused machines, fantasy animals, verbal analogies, improved designs), there is no high impact of graphic or visually concomitant imagination on general creativity. If this result is extended to teaching scenarios in preschool and middle school children's education, it is possible to suggest that perhaps many graphic or drawing activities, which the teacher asks children to do, have less than expected results or are perhaps redundant in the long run.

Neither is the association of drawing directly relevant to creative imagination. These results suggest that creative imagination is not an act of *reproduction* [49,51], but a mental function that involves elaboration and styles of thought [52], and that part of concrete images, when strengthened with external stimuli—which, in this case, is represented by the given set of imaginative exercise—allows production of something novel, something that was reflected in specific actions and involved activities of an analytical character like writing.

## 16. Conclusions

The *PIC* appears to be a suitable instrument for detecting variations in general creativity. The results point out that not only can there be variations in specific aspects of creativity, but also that general creativity can be modified stochastically or otherwise, thus challenging the theoretical consideration of Ausubel et al. (1998), who proposed that multiple intrinsic or extrinsic factors, in their isolated or combined engagements, do not have a greater incidence on creativity. Though global in its cross-cultural applications, the proposal for incorporating imaginary worlds in children's curricula for children in Latin American contexts, such as Bronstein or Bruner suggested, may not be an ineffectual contingency.

The results converge with some of the objectives of the research. The implementation of a program that develops divergent thinking will favorably influence the creativity of students. It shows that a creativity development program represents an instrument for teachers and children that provides tasks and materials with which to rehearse a variety of ways to express their creative potential. These results also suggest that the creative imagination is not an act of representation or reproduction, but a mental function that involves thought, which starts from concrete images, and is strengthened with external stimuli: In this case, the process is borne out in the set of imaginative strategies that allowed us to effectuate something new. The effects were reflected in specific actions that involved analytical activities such as writing. Finally, it should be noted that the *PIC* proved to be an adequate instrument for measurement of variations in general creativity. Even though no pedagogical adaptations have been made in the Mexican social context, this study offers evidence that allows its use to be relevant in educational practice.

**Author Contributions:** Conceptualization, J.G.G. methodology, J.G.G. and T.P.M.; validation J.G.G. and T.P.M.; formal analysis, J.G.G. and T.P.M.; investigation, J.G.G.; resources, J.G.G.; data healing, J.G.G. and T.P.M.; writing-preparation of the original draft, J.G.G. and T.P.M.; writing-revision and editing, J.G.G. and T.P.M.; visualization, J.G.G.; supervision, J.G.G. and T.P.M. project management, J.G.G.

**Funding:** This investigation did not receive external financing.

**Conflicts of Interest:** The authors declare no conflict of interest.

## Appendix A

**Table A1.** Correlation indicators for narrative, graphic, and general creativity.

| Indicators | | Fluidity | Flexib. | Orignlty | Narrative | Graphic Orignlty. | Elabor Graph. | Graphic | General |
|---|---|---|---|---|---|---|---|---|---|
| FLUIDITY | Pearson Correlation | 1 | 0.857(**) | 0.832(**) | 0.967(**) | −0.041 | 0.164 | 0.071 | 0.963(**) |
| | Sig. (bilateral) | | 0.000 | 0.000 | 0.000 | 0.835 | 0.405 | 0.718 | 0.000 |
| | N | 300 | 300 | 300 | 300 | 300 | 300 | 300 | 300 |
| FLEXIBILITY | Pearson Correlation | 0.857(**) | 1 | 0.754(**) | 0.911(**) | −0.041 | 0.048 | −0.001 | 0.901(**) |
| | Sig. (bilateral) | 0.000 | - | 0.000 | 0.000 | 0.835 | 0.808 | 0.996 | 0.000 |
| | N | 300 | 300 | 300 | 300 | 300 | 300 | 300 | 300 |
| ORIGINALITY | Pearson Correlation | 0.832(**) | 0.754(**) | 1 | 0.922(**) | 0.024 | 0.021 | 0.031 | 0.918(**) |
| | Sig. (bilateral) | 0.000 | 0.000 | - | 0.000 | 0.904 | 0.914 | 0.875 | 0.000 |
| | N | 300 | 300 | 300 | 300 | 300 | 300 | 300 | 300 |
| TOTAL NARRATIVE | Pearson Correlation | 0.967(**) | 0.911(**) | 0.922(**) | 1 | −0.038 | 0.110 | 0.041 | 0.994(**) |
| | Sig. (bilateral) | 0.000 | 0.000 | 0.000 | - | 0.849 | 0.576 | 0.837 | 0.000 |
| | N | 300 | 300 | 300 | 300 | 300 | 300 | 300 | 300 |
| ORIGINALITY GRAPHIC | Pearson Correlation | −0.041 | −0.041 | 0.024 | −0.038 | 1 | 0.058 | 0.782(**) | 0.044 |
| | Sig. (bilateral) | 0.835 | 0.835 | 0.904 | 0.849 | - | 0.770 | 0.000 | 0.824 |
| | N | 300 | 300 | 300 | 300 | 300 | 300 | 300 | 300 |
| ELABORATION GRAPHIC | Pearson Correlation | 0.164 | 0.048 | 0.021 | 0.110 | 0.058 | 1 | 0.667(**) | 0.164 |
| | Sig. (bilateral) | 0.405 | 0.808 | 0.914 | 0.576 | 0.770 | - | 0.000 | 0.404 |
| | N | 300 | 300 | 300 | 300 | 300 | 300 | 300 | 300 |
| TOTAL GRAPHIC | Pearson Correlation | 0.071 | −0.001 | 0.031 | 0.041 | 0.782(**) | 0.667(**) | 1 | 0.135 |
| | Sig. (bilateral) | 0.718 | 0.996 | 0.875 | 0.837 | 0.000 | 0.000 | - | 0.493 |
| | N | 300 | 300 | 300 | 300 | 300 | 300 | 300 | 300 |
| CREATIVITY GENERAL | Pearson Correlation | 0.963(**) | 0.901(**) | 0.918(**) | 0.994(**) | 0.044 | 0.164 | 0.135 | 1 |
| | Sig. (bilateral) | 0.000 | 0.000 | 0.000 | 0.000 | 0.824 | 0.404 | 0.493 | - |
| | N | 300 | 300 | 300 | 300 | 300 | 300 | 300 | 300 |

Note: ** Correlation is significant at the 0.01 level (bilateral).

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
