# Peer review of "The Role and Efficacy of Creative Imagination in Concept Formation: A Study of Variables for Children in Primary School"

_education, doi:10.3390/educsci9030175_

Round 1
Reviewer 1 Report
COMMENTS AND SUGGESTIONS:
Comment 1: According to template, abstract needs to have 200 words maximum. In this version of paper, abstract has 234 words. Try to reduce number of words
Comment 2: References should be numbered in order of appearance and indicated by a numeral in square brackets
Comment 3: Chapter References - references must be numbered in order of appearance in the text. Also is necessary to form all references according to the template (see Template document)
Comment 4: All chapters need to have numbers!
Wrong: The Quantitative Framework
Correct: 1. The Quantitative Framework
Comment 5: After keywords, paper starts with the sentence - "Creative Imagination appears to be active…" There is no chapter title!
Comment 6: page 11- The table name is missing above the table
Comment 7: page 11 – in first column of table
Wrong: TÃtle
Correct: Title
Comment 8: page 11 – in second column of table
Wrong: Grupo
Correct: Group
Comment 9: page 11 – in third column of table
Wrong: Media
Correct: Median or Mean
Comment 10: page 11 – row 9 in text
Written: (annexed)
Proposal: (Appendix 1 or Appendix A)
Comment 11: page 22
Written: Anexe 1
Proposal: Appendix 1 or Appendix A
Comment 12: page 22
Wrong: Correlation Indicators for Narrative, Graphic and General Creativity
Correct: Table 6. Correlation Indicators for Narrative, Graphic and General Creativity
Comment 13: page 22 - in first column of table is necessary to use English words (please translate on English)
Comment 14: page 22 – the table is not readable - increase the width of the columns
Comment 15: page 13- Table 3 is missing
Comment 16: page 15- The table name should be above the table
Comment 17: page 16- The table name should be above the table
Comment 18: page 15
Wrong: Figure 3: General Creativity Post - test
Correct: Figure 3. General Creativity Post - test
Comment 19: page 15
Wrong: Table 4: Correlation Creativity Narrative to General Creativity
Correct: Table 4. Correlation Creativity Narrative to General Creativity
Comment 20: page 15 - row 9 in text
Written: (Annex 1)
Proposal: (Appendix 1 or Appendix A)
Comment 21: page 15 - row 6 in text
Written: (see attached Table 4)
Proposal: (see Table 4)
Comment 22: page 16
Wrong: Table 5: Correlation of Graphic Creativity to General Creativity
Correct: Table 5. Correlation of Graphic Creativity to General Creativity
Comment 23: page 14
Wrong: Figure 1: Narrative Creativity
Correct: Figure 1. Narrative Creativity
Comment 24: page 14
Wrong: Figure 2: Graphic Creativity (Post – test)
Correct: Figure 2. Graphic Creativity (Post – test)
Comment 25: page 13 – in first column of table
Wrong: TÃtle
Correct: Title
Comment 26: page 13 – in second column of table
Wrong: Grupo
Correct: Group
Comment 27: page 12 – in first column of table
Wrong: TÃtle
Correct: Title
Comment 28: page 12
Wrong: Table 1. Mean and Standard deviation calculated…
Correct: Table 1. Mean and Standard deviation calculated…
Comment 29: page 13
Wrong: Table 2. Standard Deviation in Post – Test variables…
Correct: Table 2. Standard Deviation in Post – Test variables…
Author Response
Dear Reviewer
We have followed your instructions strictly and specifically, adding a few more very minor modifications or corrections which we hope you would approve.
Here is a detailed response to your suggestions:
Comment 1: According to template, abstract needs to have 200 words maximum. In this version of paper, abstract has 234 words. Try to reduce number of words
NUMBER OF WORDS HAVE BEEN REDUCED TO 200
Comment 2: References should be numbered in order of appearance and indicated by a numeral in square brackets
REFERENCES HAVE NUMBERED WITH SQ. BRACKETS
Comment 3: Chapter References - references must be numbered in order of appearance in the text. Also is necessary to form all references according to the template (see Template document)
REFERENCES HAVE BEEN NUMBERED ACCORDING TO ORDER OF APPEARANCE IN TEXT
Comment 4: All chapters need to have numbers!
Wrong: The Quantitative Framework
Correct: 1. The Quantitative Framework
ALL CHAPTERS HAVE BEEN NUMBERED
Comment 5: After keywords, paper starts with the sentence - "Creative Imagination appears to be active…" There is no chapter title!
TITLE ADDED
Comment 6: page 11- The table name is missing above the table
TABLE NAMES ARE NOW PLACED ABOVE
Comment 7: page 11 – in first column of table
Wrong: TÃtle
Correct: Title
CORRECTED
Comment 8: page 11 – in second column of table
Wrong: Grupo
Correct: Group
CORRECTED
Comment 9: page 11 – in third column of table
Wrong: Media
Correct: Median or Mean
CORRECTED
Comment 10: page 11 – row 9 in text
Written: (annexed)
Proposal: (Appendix 1 or Appendix A)
APPENDIX 1 INSERTED
Comment 11: page 22
Written: Anexe 1
Proposal: Appendix 1 or Appendix A
APPENDIX 1 INSERTED
Comment 12: page 22
Wrong: Correlation Indicators for Narrative, Graphic and General Creativity
Correct: Table 6. Correlation Indicators for Narrative, Graphic and General Creativity
CORRECTED
Comment 13: page 22 - in first column of table is necessary to use English words (please translate on English)
USED ENGLISH WORDS FOR ENTIRE COLUMN
Comment 14: page 22 – the table is not readable - increase the width of the columns
THE WIDTH HAS BEEN INCREASED to 16.5 cms
Comment 15: page 13- Table 3 is missing
CORRECTED
Comment 16: page 15- The table name should be above the table
TABLE NAME GIVEN ABOVE
Comment 17: page 16- The table name should be above the table
TABLE NAME GIVEN ABOVE
Comment 18: page 15
Wrong: Figure 3: General Creativity Post - test
Correct: Figure 3. General Creativity Post - test
CORRECTED
Comment 19: page 15
Wrong: Table 4: Correlation Creativity Narrative to General Creativity
Correct: Table 4. Correlation Creativity Narrative to General Creativity
Comment 20: page 15 - row 9 in text
Written: (Annex 1)
Proposal: (Appendix 1 or Appendix A)
Comment 21: page 15 - row 6 in text
Written: (see attached Table 4)
Proposal: (see Table 4)
'ATTACHED' ERASED
Comment 22: page 16
Wrong: Table 5: Correlation of Graphic Creativity to General Creativity
Correct: Table 5. Correlation of Graphic Creativity to General Creativity
CORRECTED
Comment 23: page 14
Wrong: Figure 1: Narrative Creativity
Correct: Figure 1. Narrative Creativity
CORRECTED
Comment 24: page 14
Wrong: Figure 2: Graphic Creativity (Post – test)
Correct: Figure 2. Graphic Creativity (Post – test)
CORRECTED
Comment 25: page 13 – in first column of table
Wrong: TÃtle
Correct: Title
CORRECTED
Comment 26: page 13 – in second column of table
Wrong: Grupo
Correct: Group
CORRECTED
Comment 27: page 12 – in first column of table
Wrong: TÃtle
Correct: Title
CORRECTED
Comment 28: page 12
Wrong: Table 1. Mean and Standard deviation calculated…
Correct: Table 1. Mean and Standard deviation calculated…
Comment 29: page 13
Wrong: Table 2. Standard Deviation in Post – Test variables…
Correct: Table 2. Standard Deviation in Post – Test variables…
Comment 11: page 22
Written: Anexe 1
Proposal: Appendix 1 or Appendix A
Comment 12: page 22
Wrong: Correlation Indicators for Narrative, Graphic and General Creativity
Correct: Table 6. Correlation Indicators for Narrative, Graphic and General Creativity
CORRECTED
Comment 13: page 22 - in first column of table is necessary to use English words (please translate on English)
CORRECTED
Comment 14: page 22 – the table is not readable - increase the width of the columns
CORRECTED
Comment 15: page 13- Table 3 is missing
ADDED
Comment 16: page 15- The table name should be above the table
ADDED ABOVE
Comment 17: page 16- The table name should be above the table
ADDED ABOVE
Comment 18: page 15
Wrong: Figure 3: General Creativity Post - test
Correct: Figure 3. General Creativity Post - test
CORRECTED
Comment 19: page 15
Wrong: Table 4: Correlation Creativity Narrative to General Creativity
Correct: Table 4. Correlation Creativity Narrative to General Creativity
CORRECTED
Comment 20: page 15 - row 9 in text
Written: (Annex 1)
Proposal: (Appendix 1 or Appendix A)
Comment 21: page 15 - row 6 in text
Written: (see attached Table 4)
Proposal: (see Table 4)
CORRECTED
Comment 22: page 16
Wrong: Table 5: Correlation of Graphic Creativity to General Creativity
Correct: Table 5. Correlation of Graphic Creativity to General Creativity
CORRECTED
Comment 23: page 14
Wrong: Figure 1: Narrative Creativity
Correct: Figure 1. Narrative Creativity
CORRECTED
Comment 24: page 14
Wrong: Figure 2: Graphic Creativity (Post – test)
Correct: Figure 2. Graphic Creativity (Post – test)
CORRECTED
Comment 25: page 13 – in first column of table
Wrong: TÃtle
Correct: Title
CORRECTED
Comment 26: page 13 – in second column of table
Wrong: Grupo
Correct: Group
CORRECTED
Comment 27: page 12 – in first column of table
Wrong: TÃtle
Correct: Title
CORRECTED
Comment 28: page 12
Wrong: Table 1. Mean and Standard deviation calculated…
Correct: Table 1. Mean and Standard deviation calculated…
CORRECTED
Comment 29: page 13
Wrong: Table 2. Standard Deviation in Post – Test variables…
Correct: Table 2. Standard Deviation in Post – Test variables…
CORRECTED
Reviewer 2 Report
In the summary you must indicate the number of participants, the number of boys and girls who participated in the study.
Present the objectives of the study more clearly
It is suggested that the authors discuss the most current study results for the years 2012, 2013, 2014, 2105, 2016, 2017, 2018 and 2019.
Discuss these results with the results of
(López 2010; Zacatelo et al. 2013; Lopéz-Martinez 2017, López (2015), Perez and Avila (2014)
The conclusions must respond more clearly than they do to whether the objectives of the study have been achieved and to what extent they have been achieved.
Unify the bibliographical references. The authors must comply with the journal publication standards
Artola, T., Ancillo, I., Barraca, J., Mosteiro, P. y Pina, J. (2004). PIC, Prueba de Imaginación 4 Creativa.Madrid: TEA Ediciones. 5
ARTOLA, T.; ANCILLO, I.; BARRACA, J.; MOSTEIRO, P. (2004, 2012). Manual de La prueba de 6 imaginación creativa. Madrid: TEA Ediciones.
The reading of Annex 1 must be in English
Author Response
In the summary you must indicate the number of participants, the number of boys and girls who participated in the study.
Present the objectives of the study more clearly
We added a section on objectives of research
It is suggested that the authors discuss the most current study results for the years 2012, 2013, 2014, 2105, 2016, 2017, 2018 and 2019.
Discuss these results with the results of
(López 2010; Zacatelo et al. 2013; Lopéz-Martinez 2017, López (2015), Perez and Avila (2014)
References have been integrated in discussions of results, page 18
The conclusions must respond more clearly than they do to whether the objectives of the study have been achieved and to what extent they have been achieved.
Conclusion has been modified
Unify the bibliographical references. The authors must comply with the journal publication standards
Bibliographical information corrected
Artola, T., Ancillo, I., Barraca, J., Mosteiro, P. y Pina, J. (2004). PIC, Prueba de Imaginación 4 Creativa.Madrid: TEA Ediciones. 5
ARTOLA, T.; ANCILLO, I.; BARRACA, J.; MOSTEIRO, P. (2004, 2012). Manual de La prueba de 6 imaginación creativa. Madrid: TEA Ediciones.
The reading of Annex 1 must be in English
Appendix 1 as per reviewer 1 suggestion added and translated into English